# Eroded telomeres are rearranged in quiescent fission yeast cells through duplications of subtelomeric sequences

Laetitia Maestroni[1], Julien Audry[1], Samah Matmati[1], Benoit Arcangioli[2], Vincent Géli [1] & Stéphane Coulon[1]

While the mechanisms of telomere maintenance has been investigated in dividing cells, little is known about the stability of telomeres in quiescent cells and how dysfunctional telomeres are processed in non-proliferating cells. Here we examine the stability of telomeres in quiescent cells using fission yeast. While wild type telomeres are stable in quiescence, we observe that eroded telomeres were highly rearranged during quiescence in telomerase minus cells. These rearrangements depend on homologous recombination (HR) and correspond to duplications of subtelomeric regions. HR is initiated at newly identified subtelomeric homologous repeated sequences (HRS). We further show that TERRA (Telomeric Repeat-containing RNA) is increased in post-mitotic cells with short telomeres and correlates with telomere rearrangements. Finally, we demonstrate that rearranged telomeres prevent cells to exit properly from quiescence. Taken together, we describe in fission yeast a mode of telomere repair mechanism specific to post-mitotic cells that is likely promoted by transcription.

[1] Marseille Cancer Research Center (CRCM), CNRS, INSERM, Aix Marseille Univ, Institut Paoli-Calmettes, Equipe labélisée Ligue contre le cancer, 13273 Marseille, France. [2] Dynamics of the Genome, UMR 3225 Genomes & Genetics; Institut Pasteur, 75015 Paris, France. Correspondence and requests for materials should be addressed to V.G. (email: vincent.geli@inserm.fr) or to S.C. (email: stephane.coulon@inserm.fr)

Telomeres are nucleoprotein structures that protect chromosomes ends from degradation and ensure replication of terminal DNA. They are composed of long stretches of double-stranded TTAGGG repeats that end with a 3′ single-stranded overhang. Because conventional replication of linear ends by DNA polymerase leads to loss of telomeric sequences, telomeres shorten progressively at each cell cycle division and this shortening triggers replicative senescence[1]. This phenomenon participates to cellular aging by limiting the proliferative capacity of most cells in an organism[2]. By contrast embryonic stem cells and germinal cells have an unlimited capacity to divide to ensure

tissue renewal, regeneration and repair. To counteract telomere attrition, these stem cells use a reverse transcriptase named telomerase that extends the 3′ end of chromosomes ends thanks to its RNA associated template[3]. However, with the exception of embryonic stem cells and germinal cells, in the majority of stem cells the telomerase activity is low or absent[4]. Thus telomere shortening occurs during replicative aging in stem cells, possibly at a slower rate than that in normal somatic cells and may alter stem cells function[5]. Indeed, the capacity of stem cells to enter and exit quiescence is imperative to tissue homeostasis and to the response to life-threatening challenges[6].

Quiescence is a common life form for the cell. Indeed the majority of the cells in adult human body tissues and organs are non-dividing postmitotic cells. Although telomeres attrition is correlated with cell division, telomere shortening has been also observed in somatic cells of brain regions or skeletal muscle, regardless of their replicative activity[7,8]. This suggests that other factors than chronological cell division may cause telomeres shortening. Because telomeres are G-rich, they might be extremely sensitive to oxidative stress. Indeed, guanines can be modified to 8-oxyguanosine by reactive oxygen species (ROS)[9]. Thus, base alteration by oxidative stress or other DNA damage may alter the binding of telomeric protein, trigger DNA repair and accelerate the telomere shortening[10–12]. Whether it concerns postmitotic cells or quiescent stem cells, the observations above raise the question of how telomeres are maintained in quiescence and how the replicative senescence will impact on cell ability to enter and exit quiescence.

*Schizosaccharomyces pombe* is a key model organism since it has a high level of conservation of telomeric proteins with mammalian cells[13] and cells can be easily maintained in quiescence state by nitrogen starvation[14]. Moreover, genetic is facilitated in *S. pombe* since it has only three large chromosomes (Chr I, II, and III with the smallest one (Chr III) containing rDNA). Telomeric repeats consist in 300 bp of the degenerated sequences $G_{2-6}TTAC[A]$ while subtelomeres contain a mosaic of multiple segments that span ~50 kb of the telomere proximal-site that define subtelomeric-elements 1 (STE1), 2 (STE2), and 3 (STE3)[15]. Subtelomeres are heterochromatinized regions in which the methylation of the lysine 9 of the histone 3 (H3K9me) serves as a binding site for the heterochromatin protein 1 (HP1[Swi6]). In spite of their silenced heterochromatin status, telomeres are transcribed in long non-coding RNA named TERRA[16]. Transcription of TERRA starts within the adjacent subtelomeric sequences and contains a variable number of telomeric G-rich sequences. In addition to TERRA, fission yeast chromosome ends produce a variety of telomeric transcripts[16,17]. Although telomeric transcription functions are extensively investigated, the role of TERRA in the biology of telomeres is still elusive.

As in mammals, telomeres from yeast cells with reduced telomerase activity progressively shorten with each cell division. At late time point of senescence, most of the cells die or remain arrested. This arrest (or crisis) is caused by a DNA damage checkpoint that is activated directly as a result of unprotected short telomeres being recognized as irreparable double-strand breaks (DSBs). In fission yeast, cells eventually recover from growth and establish survivors after 100 generations[18]. Most cells survive by circularizing all chromosomes (type I survivors) to bypass the need of telomerase. Telomere fusions are dependent on Rad52, the Rad16[XPF]-Swi10[ERCC1] endonuclease, the single-stranded DNA-binding protein RPA, and the Srs2 and Werner/Bloom Rqh1 helicases[19]. Cells can also survive by maintaining their telomeric repeated sequences, presumably through homologous recombination (HR), type II survivors related to ALT mechanism[20]. Finally, cell can survive continually amplifying and rearranging heterochromatic sequences (type III)[21]. These survivors are called HAATI (for heterochromatin amplification-mediated and telomerase-independent) and require the heterochromatin assembly machinery, the histone methyl transferase Clr4. Most frequently the rDNA spreads from the subtelomeric regions of Chr III to the termini of all three chromosomes. In very rare cases, subtelomeric sequences (STE) spread from the subtelomeric regions of Chr I and II to multiple sites dispersed in the genome.

Although, telomeres maintenance have been extensively studied in cycling cell, rare studied have been undertaken in quiescent cells[22]. It is established now that DNA damage occurs in quiescent cells and that distinct modes of DNA repair are used by quiescent and vegetative cells[23,24]. For example, oxidative DNA damage produced by respiration is thought to be source of DNA damage[24]. To tackle this issue we examine the stability of telomeres in quiescent cells population containing variable number of telomeric DNA repeats using fission yeast as a model. To this purpose, we investigate the stability of telomeres in G0 cells in the presence and absence of telomerase. We show that eroded telomeres are rearranged during quiescence. This reorganization corresponds to duplications of subtelomeric sequences by homologous recombination, named STEEx that are promoted by transcription at telomeres. We also demonstrate that STEEx formation that occurs specifically during quiescence prevents proper re-entry into proliferative phase. Our results highlight how non-dividing cells that harbor eroded telomeres may circumvent the lack of a functional telomere protection in the absence of replication.

## Results

**Eroded telomeres are rearranged in quiescence.** Because DNA damage accumulates in G0[23,24], we wondered whether telomeric repeats might represent specific target for DNA damage and how are they maintained during quiescence. Genomic DNA of cells maintained in G0 by nitrogen starvation from 1 to 8 days were digested by *EcoR*I and subjected to Southern blot analysis (Fig. 1a). As shown in Fig. 1a, telomeric repeats from WT cells were stable during quiescence. To further study the fate of dysfunctional telomeres in G0, we deleted the RNA of the telomerase TER1 in dividing cells to trigger progressive telomere shortening and replicative senescence[25]. We first performed daily dilution of *ter1Δ* spore colonies in cycling cells and monitor their growth and telomere shortening (Fig. 1b). In this representative experiment, crisis was reached after 90–100 population doublings (pds) concomitantly with the progressive disappearance of telomeric signal. After 1, 3, 5, and 7 days of senescence, cells with different

**Fig. 1** Eroded telomeres are rearranged in quiescence. **a** Top, relative position of the restriction sites in the telomeric and subtelomeric regions of *S. pombe* chromosomes based on pNSU70. Telomeric (Telo) and subtelomeric (STE1) probes used for southern blot are represented. Bottom, genomic DNA from WT cells was digested with *EcoR*I and southern blotted. The membrane was hybridized with Telo, STE1, and chromosomic probes. R, replicative sample; M, molecular weight marker; 1, 4, 8 correspond to the number of days in quiescence. A chromosomal probe was used as a loading control. **b** Top, freshly deleted *ter1* colony was maintained in exponential phase in liquid EMM medium for about 120 generations by serial dilutions. Population doublings were monitored by cell counting. Bottom, genomic DNA was digested with *EcoR*I and southern blotted. The membrane was hybridized with Telo and chromosomal probes. Asterisks indicate the time points of the replicative senescence kinetics that have been shifted to nitrogen-deprived medium. **c** Schematic representation of a typical senescence-quiescence coupled experiment. Cells collected at different type points of the senescence were shifted in deprived-nitrogen medium and maintained in senescence as indicated. **d** Genomic DNA from quiescent *ter1Δ* cells was digested by *EcoR*I and southern blotted. Membrane was hybridized with Telo, STE1, and chromosomal probes. 1, 4, 6, 8 correspond to number of days for which cells were in G0 before collecting. S1, S3, S5, and S7 indicate the days during the senescence kinetics at which cells were starved from nitrogen. Subtelomeric rearrangements are shown by asterisks. **e** Genomic DNA from quiescent WT and *ter1Δ* cells at S3 and S5 was spotted onto Hybond-XL membrane and hybridized with the indicated probes. Signal quantification has been performed with "Quantity one" software and STE1/Control ratio was determined

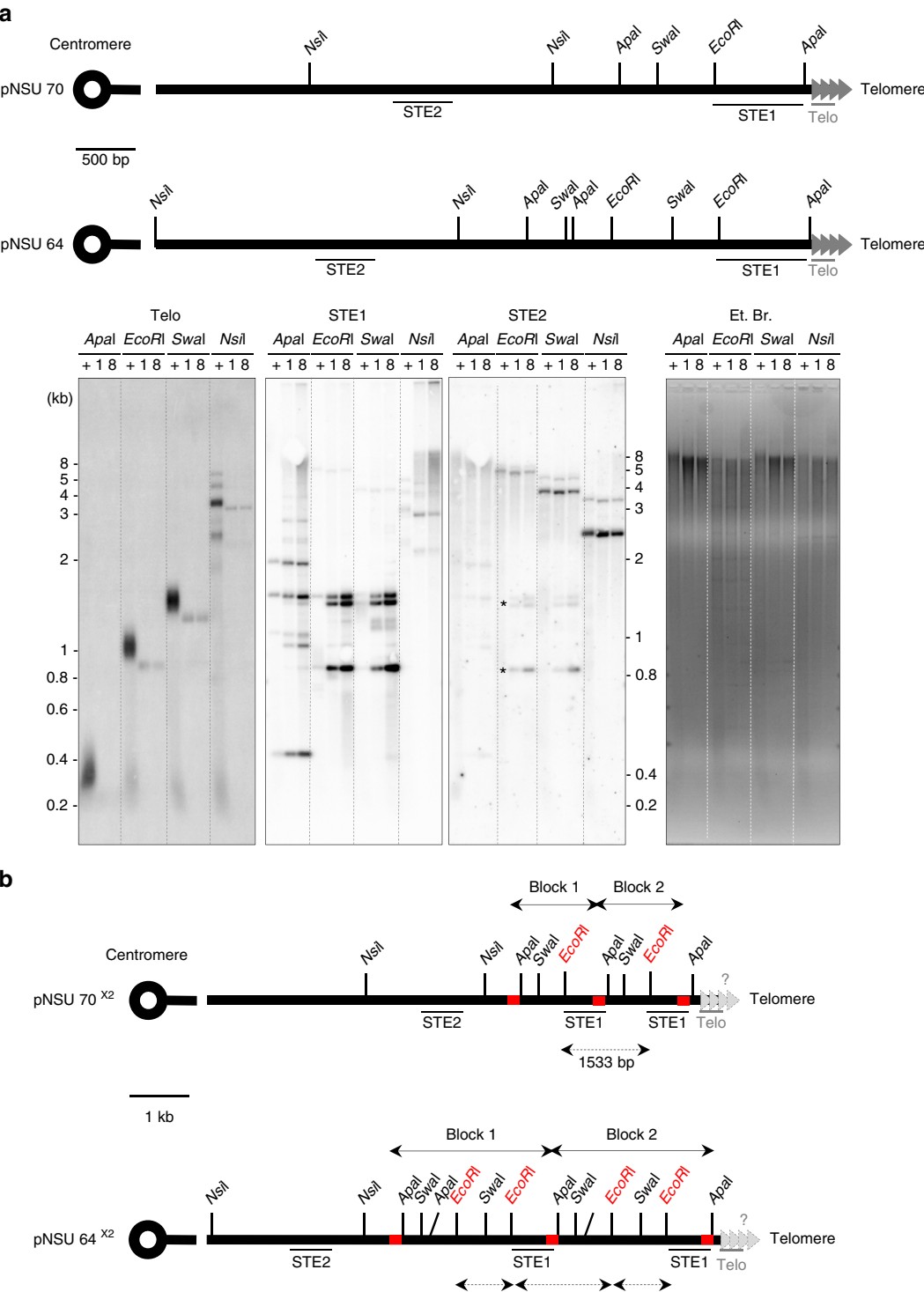

**Fig. 2** Subtelomeric rearrangements in quiescence correspond to duplication of a STE1-extended region. **a** Top, relative position of the restriction sites in the telomeric and subtelomeric regions within pNSU70 and pNSU64. Both pNSU64 and pNSU70 are representative plasmids that share significant restriction pattern similarity with pNSU65/pNSU21 and pNSU77/pNSU71, respectively[29]. The position of telomeric (Telo) and subtelomeric probes (STE1 and STE2) used for southern blot are represented. Bottom, genomic DNAs from quiescent cells (*ter1+*, after 8 days in G0 (+)) and *ter1Δ* cells (after 1 and 8 days in G0 (1 and 8)) were digested by *Apa*I, *Eco*RI, *Swa*I and *Nsi*I, and southern blotted. The membrane was hybridized with either Telo, STE1, or STE2 probes, as indicated. Ethidium bromide staining DNA was used as a loading control. Asterisks represent the residual signal from STE1 probe. **b** Schematic representation of an *Apa*I–*Apa*I subtelomeric duplication. STE1-Extended region (STEEx) within subtelomeric region of pNSU70 and pNSU64 (pNSU70$^{\times 2}$ and pNSU64$^{\times 2}$). According with this scheme, *Eco*RI digestions of pNSU64$^{\times 2}$ and pNSU70$^{\times 2}$ generate two DNA fragments of 902 and 1461 bp, and a single DNA fragment of 1533 bp, respectively. These three DNA fragments correspond to the bands that accumulated during quiescence at eroded telomeres. The presence of telomeric repeats at STEEx ends is not established

setting of telomere length (corresponding to 41, 55, 68, and 81 pds) were starved from nitrogen and maintained in quiescence for a total of 8 days (schematized in Fig. 1c). Cell counting and microscopic observations confirmed the absence of cell division during quiescence. After 1, 4, 6, and 8 days in quiescence, cells were collected and genomic DNA was extracted, digested by *EcoR*I, and subjected to southern blot analysis (Fig. 1d). The same membrane was first hybridized with a probe specific to telomeric sequences (Telo), then with a subtelomeric probe (STE1) and finally with a chromosomal probe used as a loading control. At day 1 of the senescence (S1, 41 pds) telomere size was reduced by about 120 bp compared to WT and telomeres were stable in G0 (Fig. 1d, Telo probe). However, by hybridizing the same Southern blot with the subtelomeric probe, we observed the appearance of

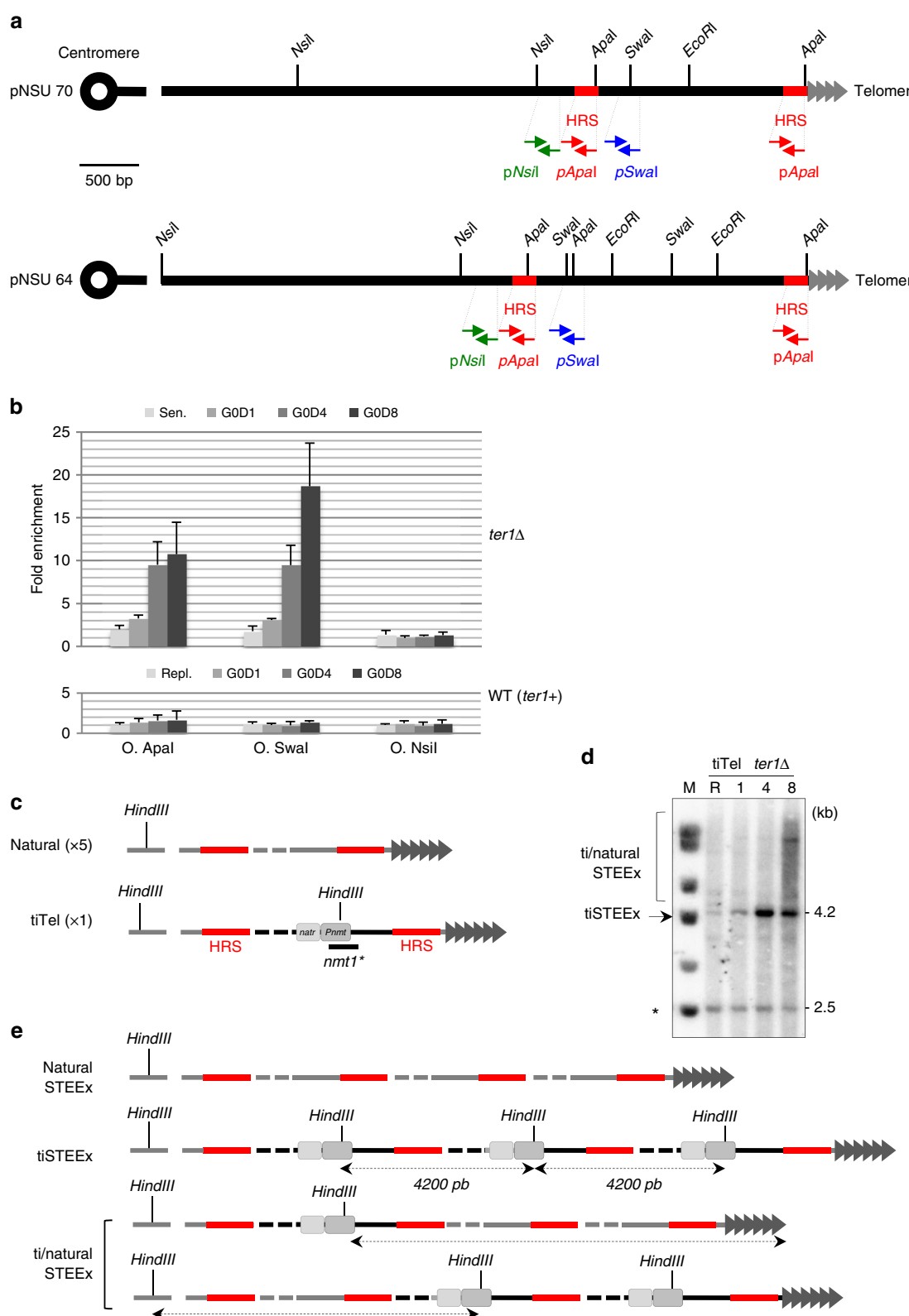

three additional bands (900, 1400, and 1500 bp) specifically in quiescence (Fig. 1d, STE1 probe) while the control chromosomal signal was constant. The subtelomeric STE1 signal was barely visible at day 1 but intensified after 4, 6, and 8 days during quiescence (Fig. 1d). When initial telomere size decreased after 3, 5, and 7 days of senescence, the appearance of STE1 signal was exacerbated. Amplification of the STE1 signal during quiescence was also confirmed by spot assays (Fig. 1e). Supplementary Fig. 1a shows another example of a ter1Δ clone for which replicative senescence and quiescence were monitored except that the Southern blot was revealed with a probe that hybridizes to both telomeric repeats and subtelomeric regions (TeloSTE1).

Finally, we wondered whether an inactive form of the telomerase could provoke the formation of these rearrangements. We took advantage of a strain expressing the catalytically dead telomerase. Subtelomeric signal amplification was observed in G0 in this strain, suggesting that the absence of telomerase activity per se induces rearrangement at eroded telomeres in quiescence (Supplementary Fig. 1b).

Collectively these results indicate that eroded telomeres are subjected to subtelomeric rearrangements in quiescence. These rearrangements increase with telomere shortening and time in quiescence. Importantly, these rearrangements occur in the total absence of cell division, without selection pressure, and at high frequency. Thus, these events are not related to survivors that occur in replicative senescence during crisis at low frequency ($10^{-4}$ to $10^{-5}$), but rather represent a new type of rearrangements that takes place globally at eroded telomeres of the quiescent cells population.

**Subtelomeric rearrangements correspond to STE1 amplification.** In the following experiments, ter1Δ cells were placed in G0 after approximately 60 pds, at a time point of senescence where rearrangements were observed. To gain insight into the mechanism by which telomere rearrangements were produced during quiescence, telomere structure from telomerase positive (8 days in G0) and ter1Δ cells (1 and 8 days in G0) were analyzed by southern blot with Telo, STE1, and STE2 probes. Schematic illustration of fission yeast subtelomeric regions cloned into plasmids (pNSU70 and pNSU64) in which probes and restriction enzyme sites are annotated, is shown in Fig. 2a. Telomeres from telomerase positive cells, maintained for 8 days in G0 (indicated by + in the southern blot of Fig. 2a), were detected as bands of ~0.3, 1, and 1.5 kb with ApaI, EcoRI, and SwaI digestion, respectively. With NsiI cut, four bands were detected that corresponds to sequence heterogeneity in subtelomeric regions as previously described[26,27]. In ter1Δ cells, telomeres were substantially shortened but still detectable after 1 and 8 days in quiescence (indicated by 1 and 8 in Fig. 2a). Digestions with EcoRI and SwaI revealed by the STE1 probe confirmed the

appearance of quiescence specific rearrangements while NsiI cut unveiled a smear at high-molecular weight, which intensified after 8 days in G0. In contrast to EcoRI and SwaI digestions, the ApaI digestion showed a mixed pattern made of a smear signal and discrete subtelomeric bands. No change in profiles was observed with the STE2 probe showing that the upstream subtelomeric STE2 region was not affected by these rearrangements. Indeed, distinct bands at 5, 4, and 2.5 kb were revealed by the STE2 probe with EcoRI, SwaI and NsiI digestions, respectively (Fig. 2a, lower panel).

Taken together, these results indicate first, that subtelomeric rearrangements occur within the terminal fragments downstream the NsiI site and second, that the amplified STE1 region contains several repetitions of DNA fragments that contain the EcoRI and SwaI restriction sites. The subtelomeric repetitions that are amplified may or not contain the ApaI site. The resulting structure of the subtelomeric amplification is schematized in Fig. 2b, which shows a representation of ApaI–ApaI subtelomeric block duplication.

**STE1 duplication are initiated at homologous sequences.** We further characterized the STE1-extended repetitions by identifying two 226 bp regions of strict homology (Homologous Repeated Sequence; HRS) flanking the subtelomeric region. One is located in the vicinity of the NsiI site while the second is adjacent to the telomeric repeats (Figs. 2b, 3a and Supplementary Fig. 2). Hence, we designed several sets of oligonucleotides indicated in Fig. 3a to map the amplified region by qPCR in the STE1-extended region. The qPCR analysis showed that G0 subtelomeric rearrangements corresponded to the Expansion of a STE1-extended DNA delineated by the ApaI-HRS sequences (named STEEx) (Fig. 3b). The results suggest that the recombination events that we observed during quiescence in cells with dysfunctional telomeres are initiated at the HRS sites in a replication-independent manner.

Furthermore, we wondered if STEEx occurs internally only in cis or also in trans with another chromosome end. To investigate this point, we deleted ter1 in a strain that contains a transcriptionally inducible telomere (called tiTel)[17]. In this strain, a natr:Pnmt1 cassette is inserted in the subtelomeric region of the left arm of chromosome II (Fig. 3c)[17]. Importantly, this modified telomere can be specifically detected using a nmt1 probe. ter1Δ tiTel cells were grown in minimal medium before nitrogen starvation. Then, STEEx formation was monitored at the tiTel telomere by southern blot with the nmt1 probe (Fig. 3d). The endogenous genomic nmt1+ sequence was visualized as a band migrating at 2.5 Kb, while the natr:Pnmt1 tiTel cassette was detected at 4.2 Kb. The tiTel signal intensified upon nitrogen starvation. As for STEEx, the amplification of tiTel signal corresponds to the duplication of HindIII-blocks that contain

**Fig. 3** Duplications of STE1-extended region are initiated at homologous repeated sequences. **a** Position of the restriction sites in the telomeric and subtelomeric regions within pNSU70 and pNSU64. Homologous repeated sequences (HRS) are marked in red. Primers within or flanking HRS that are used for quantitative PCR are represented: pApaI, pSwaI, and pNsiI. **b** Quantitative PCR analysis of subtelomeric regions in ter1+ and ter1Δ cells. Cycling ter1+ (Repl) or ter1Δ cells that were collected during replicative senescence (Sen) were shifted in deprived-nitrogen medium for 1, 4, and 8 days (G0D1, G0D4, and G0D8). Fold enrichment corresponds to the ratio of subtelomeric signal over genomic control signal (Methods section). Error bars indicate the SEM from three independent quantitative PCR. **c** Schematic representation of a natural telomere and the transcription-inducible telomere (tiTel). The natr:Pnmt1 cassette was inserted in the left arm of chromosome II. The integrated cassette contains a new HindIII restriction site, while the natural ApaI site is disrupted upon integration. Specific tiTel probe (nmt1) and HRS are indicated. **d** STEEX formation at the tiTel. tiTel ter1Δ cells were collected after 59 pds during replicative senescence and shifted to deprived-nitrogen medium. Cells were collected after the indicated days (1, 4, and 8) in quiescence. Genomic DNA was digested with HindIII and analyzed by southern blot with a nmt1 probe. R corresponds to tiTel ter1+ cells collected during replicative growth. Asterisk marks the nmt1 endogenous locus. **e** Schematic representation of natr:Pnmt1 cassette duplication (tiSTEEx). According with this scheme, HindIII-digestion of rearranged tiTel generates fragments of 4.2 Kb. The combination of tiSTEEx and natural STEEx lead to accumulation of ti/natural STEEx of heterogeneous size visualized as a smear with the nmt1 probe

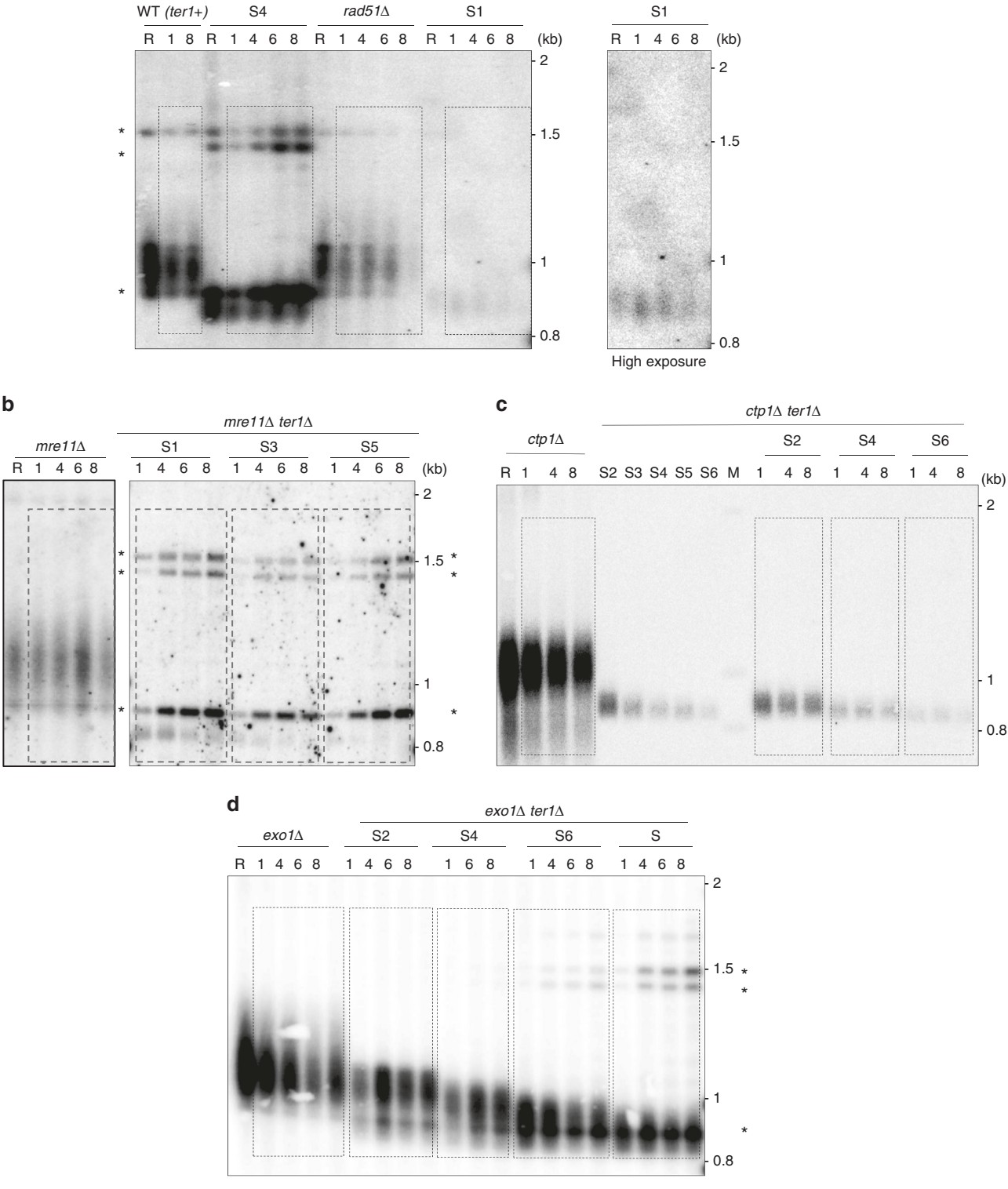

**Fig. 4** STEEx rearrangements in quiescent cells depend on Rad51 and Ctp1 but not on Mre11 or Exo1. **a–d** Telomere structure during quiescence of the indicated mutants. Genomic DNAs from single or double mutants were digested with *Eco*RI and analyzed by southern blot with a Telo/STE1 probe. *ter1+* cells and *ter1Δ* cells that were collected at different days during replicative senescence (S1–S8) and shifted to deprived-nitrogen medium. Cells were collected after the indicated days (1, 4, 6, and 8) in quiescence. R shows *ter1+* cells collected during replicative growth and M correspond to molecular weight markers. STEEx are shown by asterisks

the *nat^r:Pnmt1* cassette (tiSTEEx) (Fig. 3d). At day 8 of G0, a smear was clearly detected with the nmt1 probe, reflecting rearrangements between tiTel and natural telomeres (ti/natural-

STEEx) (Fig. 3d, e). These data show that duplication of subtelomeric sequences primarily occurs internally in *cis* and then propagates to other chromosome extremities.

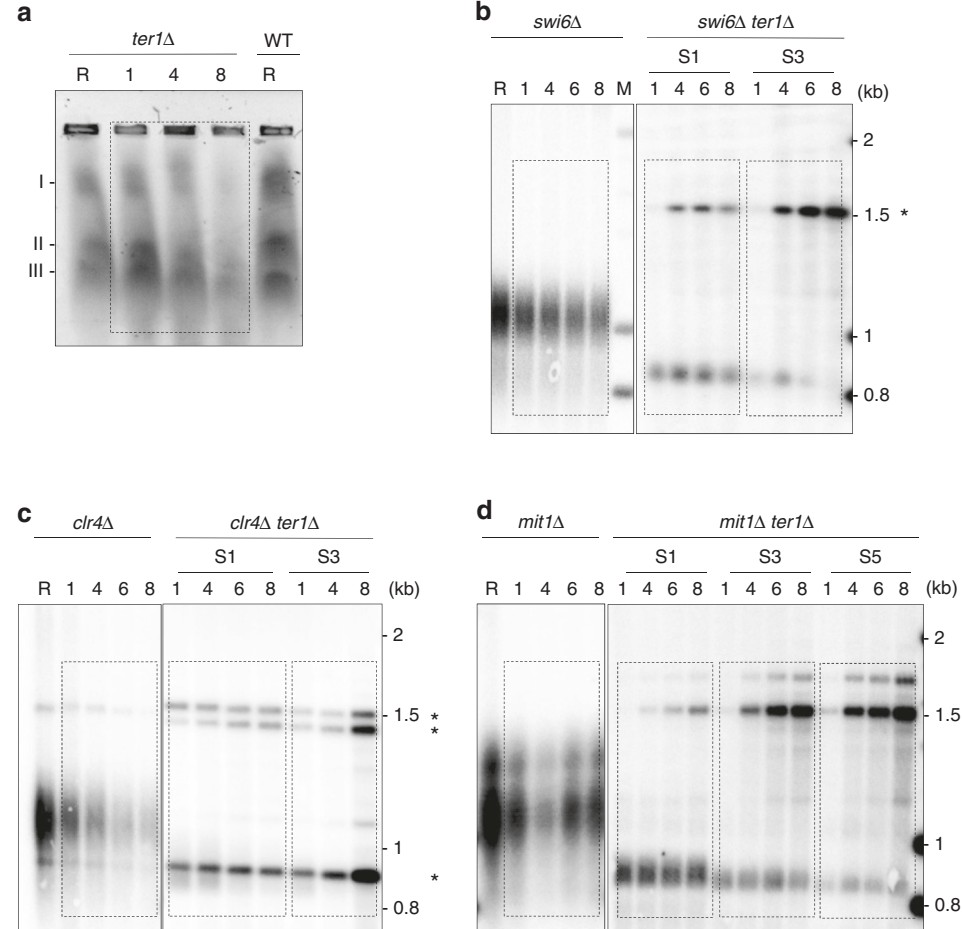

**Fig. 5** STEEx formation in quiescent cells does not depend on heterochromatin assembly machinery. **a** Analysis of STEEx chromosomes by pulsed-field gel electrophoresis. Gel was stained with ethidium bromide. **b**–**d** Genomic DNA from quiescent telomerase positive cells and *ter1Δ* cells with the indicated genetic backgrounds was digested with *EcoR*I and southern blotted. Membranes were hybridized with a Telo/STE1 probe. Cells were collected after the indicated days (1, 4, 6, and 8) in quiescence. R shows *ter1+* cells collected during replicative growth. STEEx are shown by asterisks

**Rad51 and Ctp1 are involved in STEEx formation**. The results above indicate that subtelomeric rearrangements were initiated at HRS sites through HR events. To strengthen this assumption we checked the impact of the deletion of genes involved in HR and DNA-end resection on STEEx formation (*rad51*, *mre11*, *ctp1*, and *exo1*). First, we deleted *ter1* in a *rad51Δ* background. Replicative senescence was strongly impacted by the absence of Rad51 showing the importance of HR pathway in absence of telomerase[28]. Indeed, crisis occurred shortly after the deletion of *ter1*. Nevertheless, we could starved from nitrogen these cells after 30 pds (telomere repeats were reduced by ~120–130 bp) and keep them for 8 days in quiescence. FACS analysis showed that cell death increased significantly in the *rad51Δ* strain in the presence or absence of functional telomerase. Telomere structure from *rad51Δ* and *rad51Δ ter1Δ* cells was analyzed by southern blot (Fig. 4a). In *rad51Δ* cells telomeres displayed a WT length and remained stable in G0. The signal reduction observed after 8 days in quiescence was caused by cell death. In *ter1Δ rad51Δ*, telomeres were shortened but not rearranged indicating that STEEx events did not occur. *Nsi*I digestion profile of the *rad51Δ* strain displayed a comparable profile to WT ruling out the possibility that the absence of STEEx was due to differences in the outline of the subtelomeric regions (Supplementary Fig. 3). Next, we deleted *ter1* in *mre11Δ*, *ctp1Δ* and *exo1Δ* backgrounds (Fig. 4b–d). Senescence was accelerated in *mre11Δ* and *ctp1Δ* mutants while it was not affected in *exo1Δ* strain. STEEx formation was observed

in *exo1Δ* and *mre11Δ* mutants. Noteworthy, in *ter1Δ mre11Δ* the lowered STEEx level was likely caused by a significant death of *ter1Δ mre11Δ* cells in G0 (Fig. 4b). In contrast to the situation in the *mre11Δ* mutant, STEEx did not form in the *ctp1Δ* mutant (Fig. 4c) suggesting that Ctp1 is involved in STEEx formation. From these data, we conclud that Rad51-dependant HR and Ctp1 are required to STEEx formation while Exo1 and Mre11 appear dispensable. Unexpectedly, Ctp1 was shown to function in STEEx formation independently of the MRN complex.

**STEEx are different from HAATI**. Although very rare, the spreading and amplification of subtelomeric region was observed in absence of telomerase in the HAATI[STE] survivors[21]. Hence, we wondered whether STEEx are comparable to HAATI[STE]. We investigated chromosome structure of *ter1Δ* quiescent cells by pulsed-field gel electrophoresis (PFGE) (Fig. 5a). While chromosomes of HAATI survivors fail to enter into the gel, the chromosomes of STEEx were visible after ethidium bromide staining although the signal became fuzzy after 8 days in G0. Because HAATI requires heterochromatin formation, we next examined the importance of the H3K9 methyl-transferase Clr4, Swi6 (HP1) and Mit1 (ATP-dependent DNA helicase of the SHREC complex) in STEEx formation. Senescence was accelerated in *swi6Δ* and *clr4Δ* mutants compared to WT while it was unchanged in *mit1Δ* strain. However, the absence of Clr4, Swi6 or

Mit1 did not prevent STEEx formation (Fig. 5b–d). Noteworthy *EcoR*I digestion pattern was modified in *swi6*Δ and *mit1*Δ mutants (Fig. 5b, d). Accordingly, it was previously established that deletion of *clr4* or *swi6* increased recombination among subtelomeric regions[29,30]. Indeed, *Nsi*I digestion profiles in the *swi6*Δ, *clr4*Δ and *mit1*Δ strains were different compared to WT (Supplementary Fig. 3). This heterogeneity in subtelomeric regions likely explains the different STEEx patterns in G0. From

these results, we concluded that STEEx are clearly different from HAATI^STE.

**TERRA transcription promotes STEEx formation.** In the absence of DNA replication, we wondered if transcription could initiate recombination in quiescence. Recently, it has been shown in fission yeast that among the various telomeric transcripts,

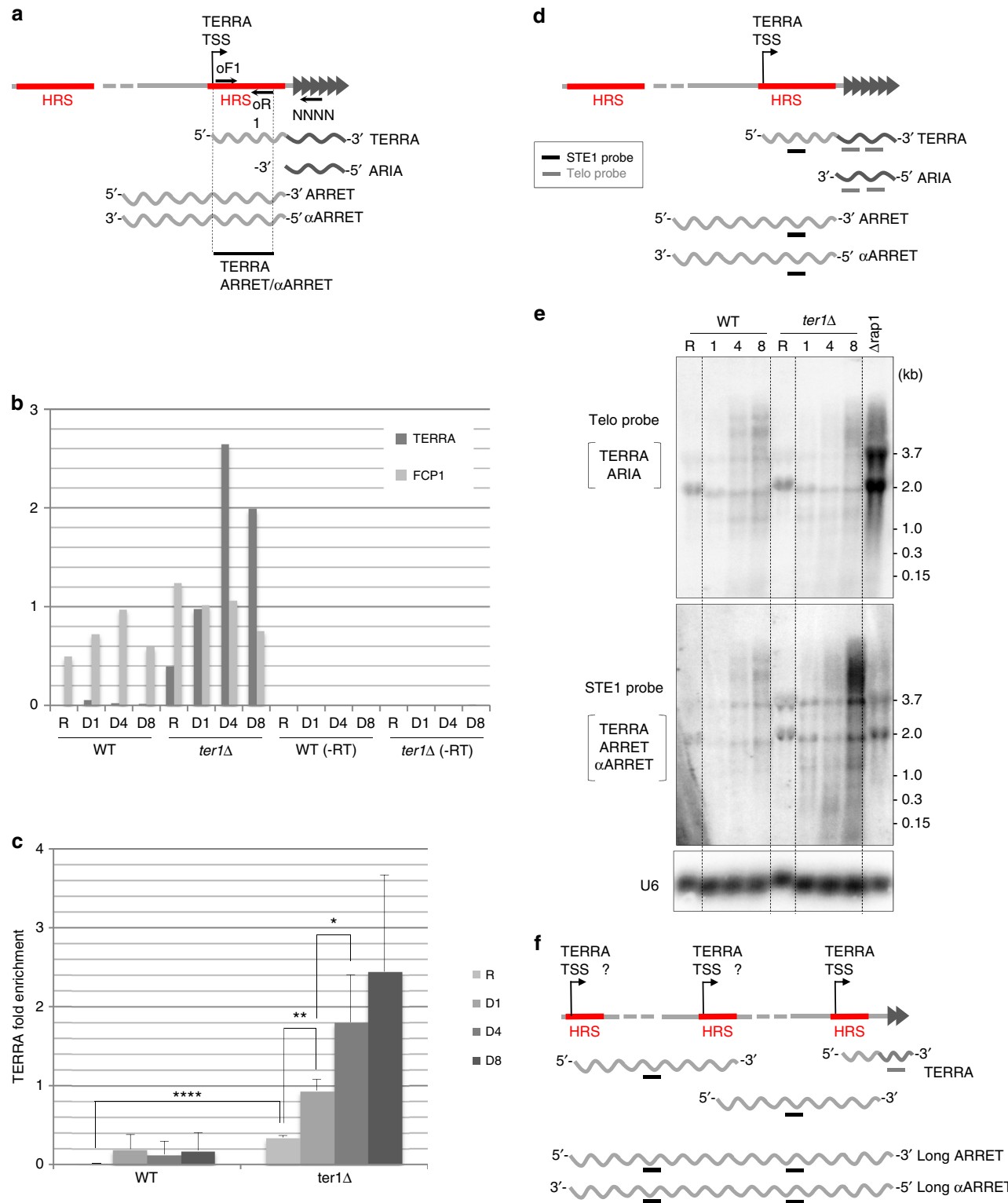

TERRA is enhanced when telomere size shortens[17]. Moreover, the observation that the HRS contains the transcription start site of TERRA (TSS) further links TERRA to STEEx. Thus, we assessed TERRA levels in quiescent cells. As an internal control we used Fcp1 mRNA that is transcribed in both cycling and quiescent cells[31]. RNA was extracted from WT and $ter1\Delta$ quiescent cells and reverse transcribed using a random hexanucleotide primer. cDNA was then subjected to real-time quantitative PCR (qPCR) with primers spanning a subtelomeric fragment downstream of the TERRA TSS (Fig. 6a). Note that the TERRA primers also amplify the ARRET and $\alpha$ARRET transcripts although they are poorly represented during senescence[17]. Figure 6b shows RNA levels of one representative experiment while Fig. 6c displays the ratio of global TERRA signal over Fcp1 signal obtained from three independent experiments. As previously reported, we confirmed that TERRA was upregulated in cycling cells when telomeres shorten. Interestingly, we observed that quiescent WT cells express TERRA at higher level than cycling cells either in the presence or in the absence of telomerase. Most importantly, we detected a significant increase of TERRA level during quiescence in $ter1\Delta$ cells, which correlated with STEEx formation.

To determine if qPCR signal amplification reflected extensive accumulation of TERRA molecules or transcription of long subtelomeric RNA (long ARRET/$\alpha$ARRET) we ascertained the RNA species in WT and $ter1\Delta$ quiescent cells by northern blot analysis. The $rap1\Delta$ mutant was used as a control because it is known to facilitate the detection of TERRA which is otherwise barely detectable in WT cells[16]. The northern blot was hybridized with a Telo probe and a STE1 probe allowing the detection of TERRA/ARIA species and global subtelomeric transcription (except ARIA), respectively (Fig. 6d). Figure 6e shows that in WT cells TERRA/ARIA accumulates in G0 (upper panel). In $ter1\Delta$ cells, TERRA/ARIA accumulation was comparable to WT. Hybridization with the STE1 probe revealed TERRA/ARRET/$\alpha$ARRET molecules (Fig. 6e, lower panel). Strikingly, we observed a strong accumulation of subtelomeric transcripts in $ter1\Delta$ cells while the signal moderately increased in WT. This accumulation of transcripts was likely caused by ARRET/$\alpha$ARRET molecules because the signal amplification was not detected with the Telo probe (Fig. 6f). We thus concluded that STEEx formation correlates with transcription of subtelomeric regions. Importantly, although subtelomeric regions are expanded in STEEx we could not detect longer transcripts than in WT or $rap1\Delta$ strains.

**RNA–DNA hybrids promote STEEx formation**. The results above suggest that transcription of subtelomeric regions correlates with STEEx formation. In budding yeast, it has been described that RNA–DNA hybrids promote recombination in pre-senescent cells at telomeres[32]. These hybrids are known to be removed by the RNase H1 and H2 enzymes[33]. We wondered if RNA–DNA hybrids generated by telomeric transcription could influence STEEx formation. We deleted $ter1$ in cells lacking the RNase H1 ($rnh1$) and investigated STEEx formation in $ter1\Delta$ $rnh1\Delta$ quiescent cells. Senescence kinetics was carried out with $ter1\Delta$ and $rnh1\Delta$ $ter1\Delta$ clones by successive streaking on YES agar plates. After streaks 1, 2, and 3 cells were grown in minimal medium, starved from nitrogen source and kept for 8 days in quiescence. Then telomere structure was analyzed by southern blot (Fig. 7). At streak 1 telomere size was comparable for both $ter1\Delta$ and $rnh1\Delta$ $ter1\Delta$ clones but STEEx appeared prematurely in $rnh1\Delta$ $ter1\Delta$ cells compared to $ter1\Delta$. At streak 2 and 3, STEEx dramatically increased with time in quiescence. These results indicate that RNA–DNA hybrids that are likely to accumulate in $ter1\Delta$ $rnh1\Delta$ cells during quiescence stimulate STEEx formation, further suggesting that telomeric transcription initiates subtelomeric rearrangements.

**STEEx are counter selected when cells exit from quiescence**. To analyze cell viability during quiescence, we micromanipulated cell after increasing period of quiescence onto rich media and monitored the proportion of cell forming colonies (CFC). The Fig. 8a shows the percentage of wild type and $ter1\Delta$ cells that are unable to form a visible colony after 1, 4, and 8 days in quiescence. This was performed after quiescence of the senescence experiment presented in Fig. 1. While almost all telomerase positive cells exit G0 (nearly 100% of CFC for the wild type), telomerase negative cells dramatically failed to exit G0 in a way that was correlated with the telomere shortening and the time in quiescence. Strikingly, >70 % of cells under conditions in which telomeres were highly rearranged were unable to exit G0 (at senescence day 7 and 8 days of G0). We further analyzed the cells that were not generating colonies and found that all of them were either enlarging or forming micro-colonies of 2–20 cells (Fig. 8a). These observations indicated that those cells were not dead in quiescence but were not capable to grow. By micro-manipulating $ter1\Delta$ cells before nitrogen starvation, we verified that the inability of telomerase negative cells to form a colony at exit of G0 was not caused by the shortening of telomeres. Indeed, senescent cells were able to form colony with high efficiency while nitrogen starvation of the same cells affected return to growth (Supplementary Fig. 4a, b). Taking together, this indicates that STEEx prevented cells to exit properly from quiescence.

To investigate this point, we monitored by Southern blot how stable were the STEEx when quiescent cells were returning to growth. We observed that STEEx were poorly maintained in proliferative cells as indicated by the reduction of the STEEx signal after 2 and 3 days of growth (Fig. 8b left panel, refer to Fig. 1c to visualize G0 profile). PFGE showed that chromosomes were still linear at exit of G0 (Fig. 8b, right panel). We analyzed the telomeric pattern of 15 individual clones that exit from 8 days of quiescence (Supplementary Fig. 4c). The common STEEx profile was lost although some reminiscent STEEx bands were still detectable in some clones. We inferred from these results that

---

**Fig. 6** TERRA accumulates in quiescence. **a** Schematic representation of the *S. pombe* telomeric transcripts. Subtelomeric region is shown as a gray line and telomeric repeats represented by dark gray triangles. Oligonucleotides used for RT-qPCR (oF1 and oR1, and random hexanucleotide NNNN), TERRA transcription start site (TSS), and HRS are indicated. **b** RNA from telomerase positive (WT) and $ter1\Delta$ cells were extracted in proliferative (R) and in quiescent (D1–D8) cells. RNA levels were determined by RT-qPCR using a random hexanucleotide primer followed by qPCR with specific oligonucleotides for TERRA and control Fcp1. A control experiment was performed without reverse transcription step (−RT). **c** TERRA fold enrichment was determined as the ratio of TERRA over Fcp1 RNA levels. Error bars indicate the SEM from three independent experiments; $P$-values are calculated from two-tailed $t$-test ($*P = 0.0355$; $**P = 0.001$; $****P < 0.001$). **d** Schematic representation of the *S. pombe* telomeric transcripts. The STE1 and Telo probes used for the northern blots are indicated by black and gray lines, respectively. The TERRA/ARIA and TERRA/ARRET/$\alpha$ARRET transcripts are detected with Telo and STE1 probes, respectively. **e** Northern blot analysis in telomerase positive (WT), $rap1\Delta$, and $ter1\Delta$ cells. RNA was extracted in proliferative (R) and in quiescent (D1, D4, and D8) cells. The Northern blot was hybridized with a Telo, STE1 and U6 (loading control) probes. **f** Schematic representation of transcription in $ter1\Delta$ quiescent cells. Duplication of TERRA TSS in STEEx may lead to accumulation of new transcripts

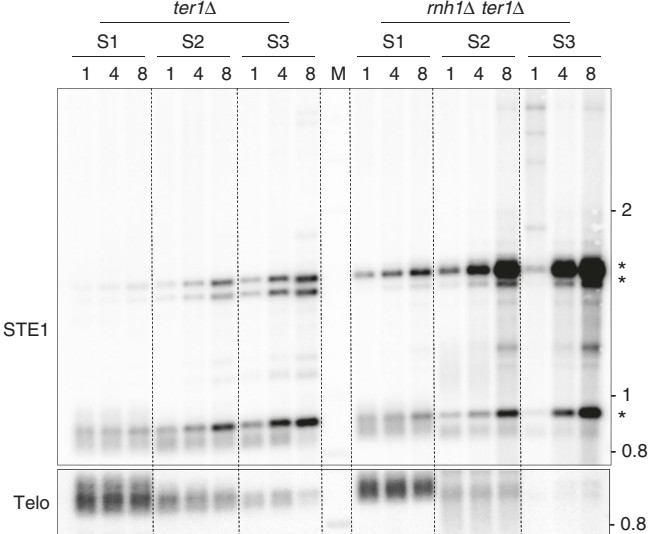

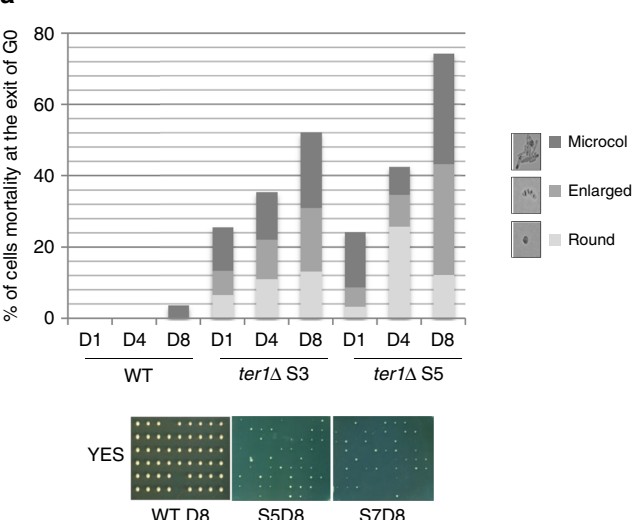

**Fig. 7** RNA:DNA hybrids promote STEEx formation. Replicative senescence (S1–S3) was carried out with *ter1Δ* and *rnh1Δ ter1Δ* cells by successive streakings on YES agar plate. Then cells were grown in minimal medium and shifted to deprived-nitrogen medium. *ter1Δ* and *rnh1Δ ter1Δ* cells were collected after the indicated days (1, 4, and 8) in quiescence. Genomic DNAs were digested with *EcoR*I and analyzed by southern blot with a Telo and STE1 probes. STEEx are shown by asterisks

cells with rearranged telomeres either died or were arrested while re-entering into the cell cycle, or that STEEx are lost when cells replicate their DNA.

## Discussion

In this study, we investigated the stability of eroded telomeres in quiescent fission yeast cells. While WT telomeres are stable, we observed that eroded telomeres in senescent cells are strongly rearranged during G0. We demonstrated that these rearrangements, named STEEx, correspond to the amplification of sub-telomeric blocks delineated by a HRS and are promoted by transcription. STEEx formation depends on Rad51-dependent HR and Ctp1 but does not seem to involve Mre11, Exo1 or the main heterochromatin assembly machinery thereby differentiating STEEx from HAATI^STE. Noteworthy, Ctp1^CTIP is known to function as co-factor of MRN complex to process DSB repair in fission yeast and human[34,35]; however, we have revealed that Ctp1 may act distinctively from Mre11 in processing eroded telomeres in quiescent fission yeast cells, likely by controlling resection. Two possible mechanisms for subtelomeric amplification are depicted in Fig. 9 (see legend for details). These mechanisms are related to BIR (Break Induced Replication) and rolling circle, depending on the usage of the HRS sequences template located in *trans* or *cis* of the eroded chromosome end. In both models, the eroded telomeres may generate ssDNA up to telomere proximal HRS. The distal HRS (located in the vicinity of *Nsi*I site) from another chromosome (BIR) or the same chromosome (rolling circle) could be used as seed to promote Rad51-dependent strand invasion, D-loop formation and DNA synthesis through migration of the D-loop. The presence of two *EcoR*I fragments of 1533 bp and 1461 bp from two chromosome ends in the WT strain (Fig. 2a) and the *Hind*III fragment of 4200 bp in the tiTel strain (Fig. 3c) supports the Rolling circle process (*cis*-amplification).

Comparable subtelomeric rearrangements have been previously described in fission yeast in the absence of Ku complex or in a mutant of Polα[29,36]; however, our study represents evidence

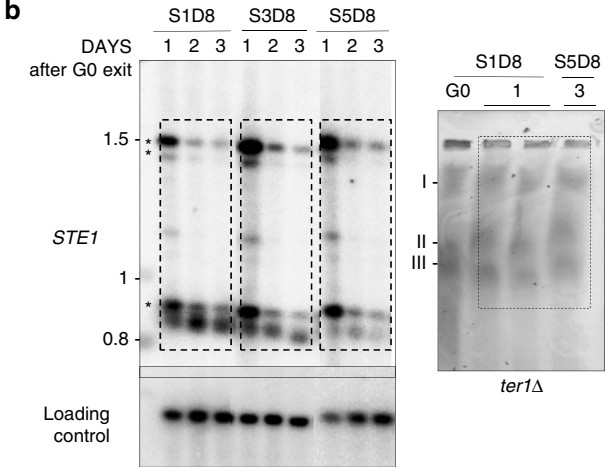

**Fig. 8** STEEx are counter selected upon exit from quiescence. **a** Wild type and *ter1Δ* single cells were maintained in quiescence for 1, 4, or 8 days, micromanipulated and plated onto YES plate allowing them to exit quiescence (refer to Fig. 1 for senescence and quiescence). The percentage of cells that were not able to form a colony was plotted. Microscopic observation was performed to determine how cells died when they re-enter into cell cycle. We distinguished single-round cell, enlarge cell, and micro-colony (1–20 cells). Round cell died in G0 while enlarged or micro-colony attempted to exit G0. **b** Left panel, at several time points of the senescence kinetics (S1, S3, and S5) a population of 8-days quiescent cells were allowed to exit G0 by changing medium to YES. Cells were grown for 1, 2, or 3 days in YES medium and collected. Genomic DNAs were prepared from the collected cultures, digested with *EcoR*I and analyzed by southern blot with a Telo/STE1 and control probes. STEEx are indicated by asterisks. Right panel, pulsed field gel electrophoresis (PFGE) analysis was performed with the indicated samples (S1D8, 1 day in senescence, 8 days in G0, and 1 day after exit; S5D8, 5 days in senescence, 8 days in G0, and 3 days of growth after exit). Gel was stained with ethidium bromide

for a massive and unique subtelomeric amplification in non-dividing conditions. Somehow, STEEx formation shares similarities with type I telomere recombination that give rise to survivors in *Saccharomyces cerevisiae* cells lacking telomerase activity. However, in contrast to budding yeast type I survivors, STEEx are not maintained when cells exit quiescence to re-enter into the cell cycle. These rearrangements probably activate the DNA damage response (as do to some extend type I telomeres in *S. cerevisiae*)

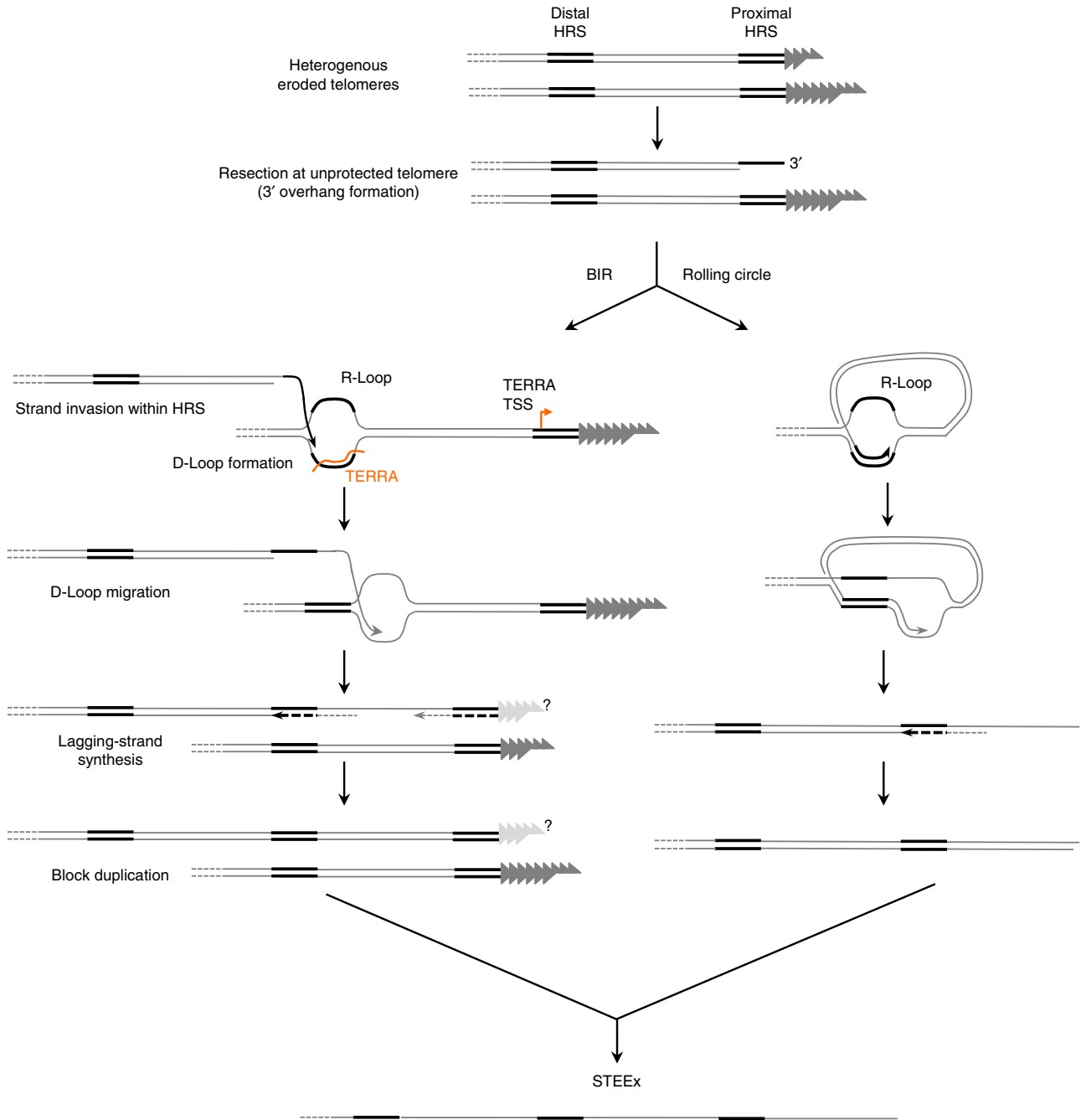

**Fig. 9** Possible mechanisms for STEEx formation in quiescence. Resection at unprotected telomeres might occur in quiescence generating a recombinogenic 3′-overhang within the telomere proximal HRS. Homology search and strand invasion by HR machinery might be facilitated by the presence of R-loop within the distal end HRS, thereby forming a D-loop structure. R-loop could be formed in subtelomeric regions by the annealing of TERRA molecules with the distal HRS. Another possibility to generate R-loop would be that RNAs are transcribed from the distal HRS that also contains the TSS sequence. Rad51-mediated HR might occur either internally within the same telomere through a rolling-circle like mechanism, or with another chromosome extremity followed by break induced repair (BIR) related process. D-loop migration and lagging-strand synthesis might then ensure duplication of subtelomeric block. Amplification of this process might lead to STEEx formation. It is not established if STEEx extremities contain telomeric repeated sequences

and are counter selected in dividing cells. We thus described in fission yeast a pathway to repair eroded telomeres that operates in telomerase minus cells and is promoted in quiescence. Recently, it was established that RNA interference (RNAi) is essential for quiescence in fission yeast, controlling heterochromatin formation at centromeres and rDNA[37]. Our work additionally shows that telomere repair in quiescence requires distinct mechanism than those that operate in cycling cells.

As schematized in the model of Fig. 9, STEEx are initiated at regions of homology (HRS) that contain the TERRA transcription site. Although quiescence is known to be characterized by global reduction of transcription[38], we have established that TERRA was induced in quiescent telomerase positive cells and that TERRA but also ARRET/αARRET transcripts was significantly increased during quiescence in telomerase negative cells (Fig. 6). Strikingly, TERRA level correlated with STEEx accumulation in G0 and

 

accumulation of RNA–DNA hybrids exacerbates STEEx formation. Thus we inferred that transcription and possibly R-loop formation at TERRA TSS could facilitate HR and STEEx formation at eroded telomeres in quiescence. On the other hand, STEEx formation could in turn exacerbate the accumulation of transcripts in quiescence. Recently, it was shown in human cells that the recruitment at telomeres of the lysine-specific demethylase 1 (LSD1) required MRE11 and was promoted by TERRA[39]. Future investigations will be required to determine whether TERRA stimulates the resection at eroded telomeres during quiescence.

STEEx was characterized by a three-band pattern that corresponds to *EcoRI*-subtelomeric DNA fragments issued from digestion of different chromosome extremities. Some of these DNA fragments can be detected at very low level during replicative senescence prior nitrogen starvation and the signal remained constant for each time point of the senescence. This suggests that STEEx formation may occur from preexisting subtelomeric regions, but are counter selected when cells proliferate. This highlights that genome stability and telomere maintenance mechanisms adapt to growth environmental conditions.

## Methods

**Strains and growth conditions**. All yeast strains used in this study are prototrophic and listed in Supplemental Table 1. Telomerase was deleted by substituting the ter1 gene by kanamycin or hygromycin cassette by one-step homologous insertion using FwTer1[275] and RevTer1[276] primers (FwTer1[275] 5′-AACG-CAACGCCCATGCTTAGAAGGTTGACAAGGAAAATTAATCAAACG GT-3′ and RevTer1[276] 5′-TTCATCTCTTCTAGTACGCAAATAAATACATTAA ATTTATTTTACATTAT-3′)[40]. Strain was grown in minimal medium MM[41], at 32 °C to a density of $6 \times 10^6$ cells per ml, washed twice in MM deprived from nitrogen (MM-N), and re-suspended in MM-N at a density of $2 \times 10^6$ cells per ml at 32 °C, reaching after two rounds of cell division $8 \times 10^6$ cells per ml[24]. This density of cell bodies is stable for several weeks and was used as a source of G0 cells. The re-entry into the vegetative cycle was performed by replacing the MM-N by one volume of fresh rich medium (YES).

**Cell viability at exit of quiescence**. Exit of quiescence was monitored by micromanipulation of single-quiescent cells on rich-medium YES plates. After 3 days of incubation at 32 °C cells colonies were counted.

**Telomere analysis and spotting assay**. Genomic DNA was prepared from $2 \times 10^8$ cells according to standard protocols and digested with the indicated restriction enzymes (New England Biolabs). The digested DNA was resolved in a 1.2% agarose gel and blotted onto a Hybond-XL membrane (GE Healthcare). After transfer, the membrane was cross-linked with UV and hybridized with different probes listed in Supplemental Table 2. [32]P labeling of DNA probes was performed by random priming using Klenow fragment exonuclease-(New England Biolabs), in presence of [α–[32]P]CTP and hybridizations were performed in Church buffer at 65 °C for Telo and subtelomeric (STE1) probes and 55 °C for chromosomal probes. For spotting assays, several dilutions of genomic DNA were directly spotted onto a Hybond-XL membrane (GE Helthcare). Radioactive signal were detected using a Biorad molecular imager FX.

**Quantitative PCR analysis**. Oligonucleotides used for qPCR: Subtelomeric primers (Fw-pNsiI[601] 5′-GCACCCTAACGCACTCAAGCCCAC-3′, Rev-pNsiI[602] 5′-GAAATGTGTGGAAGTTGAGTATGTTGGAG-3′, Fw-pSwaI[599] 5′-CGAATACT CGCCTTACGGCTCGGC-3′, Rev-pSwaI[600] 5′-GAAGATAAAAGCAGAGGAC TATATTGG-3′, Fw-pApaI[119] 5′-TATTTCTTTATTCAACTTACCGCACTTC, Rev-pApaI[252] 5′-GTGTGGAATTGAGTATGGTGAA-3′, and chromosomic primers (Fw[214] 5′-TACGCGACGAACCTTGCATAT-3′, Rev[215] 5′-TTATCAGACCA TGGAGCCCATT-3′).

**RNA isolation and analysis**. Total RNA was isolated from $2 \times 10^8$ of vegetative cells and $10 \times 10^8$ of quiescent cells, using the hot phenol method[42]. Total RNA was treated three times with RNase-free DNase I (Qiagen) and reverse transcribed using the Superscript III reverse transcriptase (Life technologies) with random hexanucleotide primers (Sigma-Aldrich). No reverse transcriptase control is used to assess DNA contamination. Quantitative PCR amplification of cDNA was carried out using the SYBR Premix Ex Taq II (Ozyme) with these cycling parameters: 1 cycle at 95 °C for 30 s, followed by 45 cycles at 95 °C for 10 s, 60 °C for 15 s and 72 °C for 20 s and a final extension step at 72 °C for 5 min. TERRA cDNA was quantified by using the primers oF1 (5′-GAAGTTCACTCAGTCATAATTAA TTGGGTAACGGAG-3′) and oR1 (5′-GGGCCCAATAGTGGGGGCATTGTATT

TGTG-3′)[17]. Fcp1 cDNA was quantified by using oFcp1-F (5′-CGAGACCATGA ACTTGAACG-3′), and oFcp1-R (5′-CATCAACGCCCAAAGGGATA-3′) primers. For northern blot analysis, RNA was electrophoresed in 1.2% formaldehyde agarose gels, transferred to Hybond-XL membrane (GE Helthcare) and hybridized to radioactively [32]P labeled probes at 42 °C (Supplementary Table 2). Oligonucleotide U6 (5′-ATGTCGCAGTGTCATCCTTG-3′) was 5′ end-labeled with T4 polynucleotide kinase (New England Biolabs) in presence of [γ–[32]P]ATP[17]. Membranes were washed twice in 0.1× Ssc, 0.1% SDS for 20 min at 42 °C. Radioactive signal was detected using a Biorad molecular imager FX.

**Pulsed-field gel electrophoresis (PFGE)**. $1 \times 10^8$ cells were resuspended in 50 µl of SEZ buffer (1 M Sorbitol, 50 mM EDTA, 0,5 mg/ml zymolase-100T) and 30 µl of agarose low gelling (Sigma-Aldrich) were added to a 0.75% final concentration. Cell suspension was transferred in one plug mold. Plugs were incubated at 37 °C during 2 h and 12 h for vegetative and quiescent cells, respectively. Plugs were rinsed 30 min in TE-1% SDS and incubated twice 12 h in Proteinase K buffer (0.5 M EDTA, 50 mM Tris-HCl, pH 9.5, 1 mg/ml Proteinase K). Plugs were rinsed three times in TE containing 1 mM PMSF for 1 h. Finally, plugs were incubated 3 h at 37 °C in 1× TAE containing 100 µg/ml of RNase. Plugs were loaded onto 0.8% agarose gel in 1× TAE buffer. Electrophoresis was performed in a CHEF DR III pulsed-field electrophoresis system (Biorad) in 1× TAE buffer with the following settings: 48 h at 2 V/cm with a 1.8 T switch time at an included angle of 106°. DNA was visualized by ethidium bromide staining (1 µg/ml) for 30 min.

**Data availability**. The authors declare that the data supporting the findings of this study are available within the paper and its Supplementary Information files and available from the corresponding author(s) upon reasonable request.

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

## Acknowledgements

We thank J. Cooper, M. Ferreira, T. Fischer, T. Nakamura, and A. Verdel for sharing biological material, in particular C. Azzalin for the gift of tiTel strain. This project was funded by the "Agence Nationale de la Recherche" (ANR-QuiescenceDNA SVSE8) and AGEMED, INSERM Aging transversal program. L.M. is supported by the ANR-QuiescenceDNA. V.G laboratory is supported by the "Ligue Nationale Contre le Cancer" (LNCC) (Equipe labélisée). S.C. is supported by "Projet Fondation ARC" (Association pour la Recherche contre le Cancer). S.M. is supported by the LNCC and J.A. was supported by the ARC.

## Author contributions

L.M. and S.C. designed and performed the experiments and analyzed the data. J.A. and S.M. performed the experiments. S.C., B.A., and V.G. designed the study. S.C. and V.G. wrote the paper.

## Additional information

**Competing interests:** The authors declare no competing financial interests.

