## [Peer Review File · Nature Communications]

Reviewers' comments:

Reviewer #1 (Remarks to the Author):

In this manuscript entitled "Eroded telomeres are rearranged in quiescent fission yeast cells through duplications of subtelomeric sequences", the authors addressed the question how dysfunctional telomeres are processed in quiescent fission yeast cells. They have shown that subtelomeric rearrangement takes place during nitrogen-starved quiescence in telomerase RNA knockout (*ter1-D*) cells but not in wild type cells. This rearrangement was named STEEx (Expansion of a STE1) and depends on *rad51* and *ctp1*. Such rearrangement has been proposed to correlate with increased telomeric repeat-containing RNA (TERRA) transcription and decreased survival rate following exit from quiescence. The author proposed that unprotected telomeres are resected in quiescence, generating a recombinogenic overhang. Increased TERRA expression facilitates strand invasion and consequently subtelomeric rearrangement. Their finding that short dysfunctional telomeres caused by *ter1* deletion are subjected to rearrangement during quiescence is a novel phenomenon, and potentially influence thinking in the field. Although their experimental flow is straightforward, important controls are missing in several experiments. At this point, this manuscript is preliminary and I cannot support the publication unless the authors address the following concerns.

Major comments:

1. I am skeptical that TERRA increases in *ter1-D* cells during quiescence and TERRA has causal relationship to STEEx. In Figure 6 b and c, the authors claim that TERRA expression level increased compared to control *fop1* expression during G0 phase. However, since HRS that contains transcription start site is amplified during G0, it is possible that amplified subtelomeric sequence give rise to long TERRA and ARRET species, which contain more than two HRS in one molecule. If this happens, qPCR result does not support author's claim. To address this, authors should perform Northern analysis. Although TERRA is hardly detectable in wt cells, a few fold increase of TERRA has been reported to be detectable (ref. 1). A deletion of *rap1* can be used as a positive control. In Figure 6e, the authors claim that overexpression of TERRA increased STEEx in *ter1-D* cells during G0 phase. However, the fold change is very subtle and not convincing. The authors should perform statistical analysis to confirm their claim. It is also important to confirm that TERRA is really expressed during G0 phase as expected.

2. In Figure 7 and S5, authors inferred that cells with rearranged telomeres either died or were arrested while re-entering into the cell cycle, or that STEEx are lost when cells replicate their DNA. However, current data set only shows correlation between STEEx and cell death after exit from G0 and causal relationship between them is not clear. I suggest to use *rad51-D*, *ctp1-D* and/or *exo1-D* background to address if STEEx causes cell death upon exit from G0. For example, comparing Figure 1d and Figure 4c suggests that telomere length of *ter1-D* at 5 days of senescence and that of *ctp1-D ter1-D* at 3 days of senescence seem to be comparable, although the authors need to confirm this on the same membrane. Yet, *ctp1-D ter1-D* cells show significantly reduced STEEx during quiescence. Thus, if STEEx has a causal link to cell death, *ctp1-D ter1-D* cells should show decreased cell death upon exit from G0 phase. Single *ctp1-D* mutant gives basal cell death ratio of this strain upon exit from G0 phase and can be used as a control to estimate a negative effect of *ctp1* deletion on cell mortality upon G0 exit.

Minor comments:

1. The term "TERRA" should be spelled out in the first place.
2. Characters of telomeric DNA (such as length and sequence) and TERRA in fission yeast should be explained in the introduction, while description about quiescence in higher eukaryotes and shelterin-like proteins are not very relevant to this study and can be shortened.

3. In line 23 of page 9, a reference of "Moravec and colleagues (2016)" is not shown properly.
4. In the last line of page 9, "Fig. 6e and S5" should be "Fig. 6e and S4".
5. In Figure 5b, the lane between swi6-D-8 and swi6-D ter1-D-S1-1 should be labeled. If it's marker lane, label as such. This also applies to Figure 6e, Figure S1a and b, and Figure S5a and c.
6. In Figure 5b, STEEx is not shown by asterisks as indicated in figure legend.
7. In Figure 1d, wt control is very important in this experiment and should be shown side by side with ter1-D results, although it is currently shown in supplementary Figure S1.

Reference

1. Fission yeast Cactin restricts telomere transcription and elongation by controlling Rap1 levels. Luca E Lorenzi, et al., the EMBO Journal (2015) 34, 115-129

Reviewer #2 (Remarks to the Author):

This study describes the fate of eroded telomeres in quiescent cells in fission yeast. Rather surprisingly, and interestingly, the authors find that shortened telomeres (achieved by eliminating telomerase) were unstable in quiescent cells. Specifically, the authors go to great lengths to demonstrate that, although telomeric repeats are stably maintained in these cells, subtelomeric elements are rearranged and amplified into configurations that they term STEEx. These rearrangements are dependent on recombination (rad51) but, unexpectedly (given the requirement for cdt1), not on mre11. Interestingly, a correlation is found between levels of telomeric RNA (TERRA) and STEEx. Finally, STEEx telomeres are found to be unstable and counter-selected upon exit from quiescence.

These findings are interesting, and significantly add to the growing awareness of the complexities of outcomes in eroded telomeres. It would be interesting to know whether these rearranged telomeres might arise as transient intermediates in normally growing wild-type cells.

Major points.

While elsewhere authors are very careful in linking TERRA production to STEEx formation, one section of the results is much less guarded: 'TERRA transcription promotes STEEx formation'. I remain unconvinced that the evidence presented clearly links TERRA to STEEx. First, the data only show that the two processes are correlated, not that there is a causal link. In this regard, the experiment shown in Figure 6e is key and would seem the most direct way to establish causality. Unfortunately, the evidence presented shows a tiny effect of induction of transcription at the Pnmt telomere, which the authors do not even attempt to quantify and it is certainly not statistically significant. Second, even the correlative evidence linking TERRA levels to STEEx is perhaps over-interpreted. The increase in transcription could be linked to further levels of telomere erosion which might be hard to detect by Southern. Or it could be due to the start of transcription near the HRS region being positioned further away from repressive telomeric repeats in the expanding STEEx clones. Or there might be multiple start sites being present in the amplified STEEx leading to increased chance of transcription. Or there might be a technical issue with longer RNAs being produced, each containing multiple copies of the target amplicons, and therefore producing a higher signal in the QPCR assay despite similar levels of RNA being present in terms of number of molecules. In short I am unconvinced about the role, if any, of TERRA, in STEEx production.

In Figure 7a, I do not understand whether the wild-type control has undergone G0 arrest or not, and, if so, for how long. A comparison of the survival rate of the telomerase mutants with wild-type is lacking in all these experiments, making them uninterpretable. It is perfectly possible that viability drops for wild-type cells as well, with prolonged arrest in G0. This experiment, like the one in Figure S5, is hard to assess in the absence of matched controls with the wild-type.

Minor points

I do not understand the curve shown in Figure 1b and how the experiment was conducted. How is it determined that crisis was reached after 90-100 generations from this?

From Figure 2a it is concluded that, unlike for subtelomeric element STE1, element STE2 did not undergo changes and therefore this region was not affected by rearrangements. Are there stronger exposures of the gels to support this conclusions? Bands shown are extremely faint in many cases. Is the signal present in the STE2 lanes due to cross-hybridization?

Reviewers' comments:

Reviewer #1 (Remarks to the Author):

In this manuscript entitled "Eroded telomeres are rearranged in quiescent fission yeast cells through duplications of subtelomeric sequences", the authors addressed the question how dysfunctional telomeres are processed in quiescent fission yeast cells. They have shown that subtelomeric rearrangement takes place during nitrogen-starved quiescence in telomerase RNA knockout (*ter1-D*) cells but not in wild type cells. This rearrangement was named STEEx (Expansion of a STE1) and depends on *rad51* and *ctp1*. Such rearrangement has been proposed to correlate with increased telomeric repeat-containing RNA (TERRA) transcription and decreased survival rate following exit from quiescence. The author proposed that unprotected telomeres are resected in quiescence, generating a recombinogenic overhang. Increased TERRA expression facilitates strand invasion and consequently subtelomeric rearrangement. Their finding that short dysfunctional telomeres caused by *ter1* deletion are subjected to rearrangement during quiescence is a novel phenomenon, and potentially influence thinking in the field. Although their experimental flow is straightforward, important controls are missing in several experiments. At this point, this manuscript is preliminary and I cannot support the publication unless the authors address the following concerns.

Major comments:

1. I am skeptical that TERRA increases in *ter1-D* cells during quiescence and TERRA has causal relationship to STEEx. In Figure 6 b and c, the authors claim that TERRA expression level increased compared to control *fop1* expression during G0 phase. However, since HRS that contains transcription start site is amplified during G0, it is possible that amplified subtelomeric sequence give rise to long TERRA and ARRET species, which contain more than two HRS in one molecule. If this happens, qPCR result does not support author's claim. To address this, authors should perform **Northern analysis**. Although TERRA is hardly detectable in wt cells, a few fold increase of TERRA has been reported to be detectable (ref. 1). A deletion of *rap1* can be used as a positive control. In Figure 6e, the authors claim that overexpression of TERRA increased STEEx in *ter1-D* cells during G0 phase. However, the fold change is very subtle and not convincing. The authors should perform statistical

analysis to confirm their claim. It is also important to confirm that TERRA is really expressed during G0 phase as expected.

We agree that the impact of overexpression of TERRA using tiTel on STEEx formation is not convincing (and statistically not relevant), although this effect is clearly reproducible. We think that the slight effect of TERRA overexpression is due to the leak of *nmt1* promoter and to fact that TERRA levels are already high in G0. Thus, we decided to remove Figures 6d and 6e (tiTel induction experiments) of the initial version of the MS.

As requested by reviewer #1, we performed Northern analysis to detect TERRA in G0 to eliminate potential erroneous interpretations of the qPCR results. RNA was extracted from WT, *rap1Δ* and *ter1Δ* cells. Northern blots were hybridized either with a telo or a STE1 probe (new Figure 6d-e, see p9-10 for details). These new results show that TERRA is present in quiescent WT cells to a level that is comparable to the one in *rap1Δ* cells. Moreover, we show that subtelomeric transcripts massively accumulate in *ter1Δ* cells. Thus, we directly confirm that transcription at telomeres is enhanced in G0 and exclude a possible bias of the qPCR. Finally these results indicate that STEEx correlate with the accumulation of specific RNA molecules, possibly TERRA / ARRET / α ARRET.

To further assess the role of transcription of subtelomeric regions in STEEx formation, we used a *rnh1Δ ter1Δ* strain and monitored the telomere structure in G0 (new Figure 7). This Figure clearly shows that deletion of the Rnh1 RNA-DNA hybrid ribonuclease greatly enhances STEEx formation, thereby linking RNA:DNA hybrid to STEEx.

2. In Figure 7 and S5, authors inferred that cells with rearranged telomeres either died or were arrested while re-entering into the cell cycle, or that STEEx are lost when cells replicate their DNA. However, current data set only shows correlation between STEEx and cell death after exit from G0 and causal relationship between them is not clear. I suggest to use *rad51-D*, *ctp1-D* and/or *exo1-D* background to address if STEEx causes cell death upon exit from G0.

For example, comparing Figure 1d and Figure 4c suggests that telomere length of *ter1-D* at 5 days of senescence and that of *ctp1-D ter1-D* at 3 days of senescence

seem to be comparable, although the authors need to confirm this on the same membrane. Yet, *ctp1-D ter1-D* cells show significantly reduced STEEx during quiescence. Thus, if STEEx has a causal link to cell death, *ctp1-D ter1-D* cells should show decreased cell death upon exit from G0 phase. Single *ctp1-D* mutant gives basal cell death ratio of this strain upon exit from G0 phase and can be used as a control to estimate a negative effect of *ctp1* deletion on cell mortality upon G0 exit.

Referee #1 proposes to monitor cell viability upon exit of G0 in a mutant that does not generate STEEx (ex. *ctp1Δ* or *rad51Δ*) in order to check if the absence of STEEx decreases cell death at exit of G0. This experiment is proposed to address the causality between cell death at exit of G0 and STEEx formation.

This issue is not easy to address because *ctp1Δ* and overall *rad51Δ* telomerase positive cells already exhibit a strong mortality in G0. Indeed, while in WT telomerase positive cells the percentage of mortality is very low (3-4%), this rate is high for *ctp1Δ* cells (above 20% or more). Note that the *mre11Δ* mutant that makes STEEx in cells lacking telomerase also dies in G0.

Nevertheless following Referee #1 suggestion, we deleted *ter1* in *ctp1Δ* cells and monitored telomere structure, cell death in G0 (determined by FACS), and capacity to exit quiescence in *ter1Δ* and *ctp1Δ ter1Δ* cells. In these experiments, telomere length of both mutants (*ter1Δ* and *ctp1Δ ter1Δ*) prior quiescence was similar (see the Figure below, upper panel, triangles indicate the day of senescence at which telomere size is comparable in *ter1Δ* and *ctp1Δ ter1Δ* cells).

After 8 days in G0, the percentage of cell mortality in G0 of *ter1Δ* and *ctp1Δ ter1Δ* cells reaches approximately 8% and 38%, respectively (see the Figure below, left lower panel). When these cells are further micromanipulated and put on a rich medium agar plate allowing them to exit from G0, the percentage of cell mortality at exit of G0 raises up to 50% and 90% for *ter1Δ* and *ctp1Δ ter1Δ*, respectively (see the Figure below, right lower panel). If the percentage of cell death in G0 (8% and 38%) is subtracted, we can theoretically infer that the rate of cells mortality at the exit of G0 is 42% for *ter1Δ* and 52% for *ctp1Δ ter1Δ*. We can estimate that 39% of *ter1Δ* cells (that generate STEEx) and 32% (or below) of *ctp1Δ ter1Δ* cells (that do not make STEEx) die at exit of G0. From these figures, it is therefore difficult to determine whether cells that do not make STEEx like *ctp1Δ ter1Δ* cells exhibit a lower rate of cell mortality at exit of G0. This point would have been better addressed with a mutant that does not exhibit an elevated level of cell death in G0. Until now, we did not find such a mutant. We have done our best to provide the information that the referee requested.

Minor comments:

1. The term "TERRA" should be spelled out in the first place.

Fixed

2. Characters of telomeric DNA (such as length and sequence) and TERRA in fission yeast should be explained in the introduction, while description about quiescence in higher eukaryotes and shelterin-like proteins are not very relevant to this study and can be shortened.

We have modified the introduction accordingly. A paragraph describing characters of telomeric DNA was added (see page 3), while the paragraph related to the fission yeast shelterin was removed

3. In line 23 of page 9, a reference of “Moravec and colleagues (2016)” is not shown properly

Fixed

4. In the last line of page 9, “Fig. 6e and S5” should be “Fig. 6e and S4”.

This paragraph has been removed

5. In Figure 5b, the lane between swi6-D-8 and swi6-D ter1-D-S1-1 should be labeled. If it's marker lane, label as such. This also applies to Figure 6e, Figure S1a and b, and Figure S5a and c.

Thank you for this remark. We have labeled the lanes accordingly.

6. In Figure 5b, STEEx is not shown by asterisks as indicated in figure legend.

Asterisks have been added.

7. In Figure 1d, wt control is very important in this experiment and should be shown side by side with ter1-D results, although it is currently shown in supplementary Figure S1

We are sorry but the WT control was not loaded on the same gel (Fig 1d) as it is presented in Fig S1. In the new version of figure 1A, we now show a Southern blot in which a WT G0 genomic sample is hybridized with Telo and STE1 probes. This confirms that STEEx are not formed in quiescence in a WT strain.

Reference

1. Fission yeast Cactin restricts telomere transcription and elongation by controlling Rap1 levels. Luca E Lorenzi, et al., the EMBO Journal (2015) 34, 115-129

Reviewer #2 (Remarks to the Author):

This study describes the fate of eroded telomeres in quiescent cells in fission yeast. Rather surprisingly, and interestingly, the authors find that shortened telomeres (achieved by eliminating telomerase) were unstable in quiescent cells. Specifically, the authors go to great lengths to demonstrate that, although telomeric repeats are stably maintained in these cells, subtelomeric elements are rearranged and amplified into configurations that they term STEEx. These rearrangements are dependent on recombination (*rad51*) but, unexpectedly (given the requirement for *cdt1*), not on *mre11*. Interestingly, a correlation is found between levels of telomeric RNA (TERRA) and STEEx. Finally, STEEx telomeres are found to be unstable and counter-selected upon exit from quiescence.

These findings are interesting, and significantly add to the growing awareness of the complexities of outcomes in eroded telomeres. It would be interesting to know whether these rearranged telomeres might arise as transient intermediates in normally growing wild-type cells.

Major points.

While elsewhere authors are very careful in linking TERRA production to STEEx formation, one section of the results is much less guarded: 'TERRA transcription promotes STEEx formation'.

I remain unconvinced that the evidence presented clearly links TERRA to STEEx. First, the data only show that the two processes are correlated, not that there is a causal link. In this regard, the experiment shown in Figure 6e is key and would seem the most direct way to establish causality. Unfortunately, the evidence presented shows a tiny effect of induction of transcription at the Pnmt telomere, which the authors do not even attempt to quantify and it is certainly not statistically significant. Second, even the correlative evidence linking TERRA levels to STEEx is perhaps over-interpreted. The increase in transcription could be linked to further levels of telomere erosion which might be hard to detect by Southern. Or it could be due to the start of transcription near the HRS region being positioned further away from repressive telomeric repeats in the expanding STEEx clones. Or there might be multiple start sites being present in the amplified STEEx leading to increased chance of transcription. Or there might be a technical issue with longer RNAs being produced, each containing

multiple copies of the target amplicons, and therefore producing a higher signal in the QPCR assay despite similar levels of RNA being present in terms of number of molecules

Please see our answer to Referee #1 point1.

In short I am unconvinced about the role, if any, of TERRA, in STEEx production.

In Figure 7a, I do not understand whether the wild-type control has undergone G0 arrest or not, and, if so, for how long. A comparison of the survival rate of the telomerase mutants with wild-type is lacking in all these experiments, making them uninterpretable. It is perfectly possible that viability drops for wild-type cells as well, with prolonged arrest in G0. This experiment, like the one in Figure S5, is hard to assess in the absence of matched controls with the wild-type.

In Figure 7a, WT (telomerase positive) cells have undergone a G0 arrest for 8 days and were then micromanipulated and put on a rich medium agar plate (exactly as *ter1Δ* cells). For sake of clarity, in the new version of the MS we show the WT control in Figure 8a (which corresponds to previous Figure 7a). As previously described (Ben Hassine et al., EMBOJ 2009), WT cells survive very well to prolonged time in G0 and exit from quiescence at a high frequency after 8 days of nitrogen starvation.

Minor points

I do not understand the curve shown in Figure 1b and how the experiment was conducted. How is it determined that crisis was reached after 90-100 generations from this?

Figure 1b is a typical "senescence curve". In details, *ter1* was freshly deleted and a *ter1Δ* clone was grown in rich medium. At each day cells were numerated, diluted and telomere length was monitored. In the absence of telomerase, telomeres shorten progressively. Telomere shortening causes telomere deprotection and activates the DDR. Generation time increases until cells do not divide anymore. Crisis corresponds to the time point at which growth rate is at its minimum. In *S. pombe*, it takes place at 90-100 generations. At this

point, some survivors may emerge and cell growth restart. We hope that these explanations will help to the understanding of Figure 1b.

From Figure 2a it is concluded that, unlike for subtelomeric element STE1, element STE2 did not undergo changes and therefore this region was not affected by rearrangements. Are there stronger exposures of the gels to support this conclusion? Bands shown are extremely faint in many cases. Is the signal present in the STE2 lanes due to cross-hybridization?

We agree that the interpretation of the STE2 southern might be misleading. To clarify this point we mention in the text (page 6) that digestion by EcoRI, SwaI and NsiI generates DNA fragments in STE2 telomeric regions of 5, 4 and 2.5 kb, respectively. The STE2 signal observed in Figure 2a is intense and does not change during quiescence. In the legend of Figure 2a, we also indicate that a residual signal from STE1 hybridization is visible (marked by an asterisk).

REVIEWERS' COMMENTS:

Reviewer #1 (Remarks to the Author):

The authors have addressed all the questions I raised and tried their best. They have added new Figure 7, in which STEEx formation is enhanced by *rnh1* deletion. The new result supports their hypothesis that TERRA/ ARRET/ aARRET molecules are involved in STEEx formation. Now I support the publication of the current manuscript in nature communications journal.

Minor points

Page 6, line187; "Fig 1a, lower panel" should be "Fig 2a, lower panel".

Fig2a lower panel, Figure 7 and Figure S3; marker size indicators on the right side of the gel are not aligned properly.

Reviewer #2 (Remarks to the Author):

I am happy with the revised version of the manuscript. The new Northern data goes some way to address my previous concerns. However, one point I was trying to make in my earlier review was that, contrary to the authors suggestion that TERRA might lead to STEEx, it could well be that STEEx might lead to higher TERRA (and ARRET etc). This remains a possibility, and the authors should explicitly mention it, even though the RNaseH result is consistent with higher RNA levels being implicated in STEEx formation. I think that this is a provocative paper which, while describing a situation unlikely to occur in yeast cells as the outcome is lethal, uncovers novel and unexpected mechanisms at play in quiescent cells.

RESPONSE TO REFEREES

REVIEWERS' COMMENTS:

Reviewer #1 (Remarks to the Author):

The authors have addressed all the questions I raised and tried their best. They have added new Figure 7, in which STEEx formation is enhanced by *rnh1* deletion. The new result supports their hypothesis that TERRA/ ARRET/ aARRET molecules are involved in STEEx formation. Now I support the publication of the current manuscript in nature communications journal.

Minor points

Page 6, line187; "Fig 1a, lower panel" should be "Fig 2a, lower panel".

This has been corrected

Fig2a lower panel, Figure 7 and Figure S3; marker size indicators on the right side of the gel are not aligned properly.

This has been modified accordingly.

Reviewer #2 (Remarks to the Author):

I am happy with the revised version of the manuscript. The new Northern data goes some way to address my previous concerns. However, one point I was trying to make in my earlier review was that, contrary to the authors suggestion that TERRA might lead to STEEx, it could well be that STEEx might lead to higher TERRA (and ARRET etc). This remains a possibility, and the authors should explicitly mention it, even though the RNaseH result is consistent with higher RNA levels being implicated in STEEx formation. I think that this is a provocative paper which, while describing a situation unlikely to occur in yeast cells as the outcome is lethal, uncovers novel and unexpected mechanisms at play in quiescent cells.

We agree with referee 2 saying that formation of STEEx may lead to higher level of TERRA / ARRET / ... Accordingly, we added this notion in the discussion: "On the other hand, STEEx formation could in turn exacerbate the accumulation of transcripts in quiescence." (p13)